# Magnitude and determinants of adequate antenatal care service utilization among mothers in Southern Ethiopia

Lielt Gebreselassie Gebrekirstos[1]*, Tsiyon Birhanu Wube[2,3], Meron Hadis Gebremedhin[4], Eyasu Alem Lake[5]

1 Department of Maternity and Reproductive Health Nursing, Wolaita Sodo University, Wolaita Sodo, Southern Ethiopia, 2 Department of Adult Health Nursing, Aksum University, Aksum, Northern Ethiopia, 3 Department of Medicine, Lazarski University, Warsaw, Poland, 4 Department of Medicine, Wolaita Sodo University, Wolaita Sodo, Southern Ethiopia, 5 Department of Pediatrics and Child Health Nursing, Wolaita Sodo University, Wolaita Sodo, Southern Ethiopia

* lieltinamulu23@gmail.com

## Abstract

### Background

Mortality from preventable pregnancy-related complications remains high in Ethiopia. Antenatal care remains a major public health intervention that prevents maternal and neonatal mortality. Thus, this study aimed to assess the magnitude and determinants of adequate antenatal care utilization in Southern Ethiopia.

### Methods

A community-based cross-sectional study was conducted between November and December 2019. A systematic random sampling technique was used to select 670 women. Data were collected using a pre-tested structured questionnaire administered with a digital survey tool (open data kit) and directly exported to STATA version 15 for analysis. Descriptive statistics followed by a multivariable logistic regression analysis were performed. Both crude and adjusted odds ratios (ORs) with 95% confidence intervals were reported.

### Results

The magnitude of adequate antenatal care utilization was 23.13%. Tertiary and above education (AOR,4.15;95%CI: 1.95, 8.83), having the best friend who used maternal care (AOR,2.01;95%CI: 1.18,3.41), husband support (AOR,3.84; 95%CI: 1.05, 14.08), high wealth index (AOR,3.61; 95%CI: 1.86, 6.99), follow-up in private health facilities (AOR, 2.27;95% CI:1.33, 3.88), having a history of risky pregnancy (AOR,2.59; 95%CI: 1.55, 4.35), and planned pregnancy (AOR,2.60;95% CI: 1.35, 4.99) were significant determinants of overall adequate ANC service utilization.

**Data Availability Statement:** All relevant data are within the manuscript and its Supporting Information files.

**Funding:** Wolaita Sodo University covers perdium for data collectors and supervisors.

**Competing interests:** The authors have declared that no competing interests exist.

**Abbreviations:** ANC, Ante Natal Care; AOR, Adjusted Odd Ratio; CI, Confidence Interval; COR, Crude Odd Ratio; WHO, World Health Organization; IMR, Infant Mortality Rate; MMR, Maternal Mortality Rate; WSU, Wolaita Soddo University.

## Conclusion

The utilization of adequate antenatal care services is quite low. The study findings suggest that interventions should be in place to improve husband's support, social networks, and women's education. There is also a need to counsel women to utilize family planning.

## Introduction

Pregnancy and childbirth-related obstetric complications are the driving causes of maternal mortality among women of reproductive age in many developing countries [1, 2]. Mortality from preventable pregnancy-related complications remains high with 300,000 maternal deaths which accounts for 66% of the global cases. Nearly 3 million babies die each year during their first month of life in low and middle-income countries. Ethiopia is one of the 24 countries with a maternal mortality ratio greater than 400 and a high infant mortality rate of 48% [3–7]. Most maternal and fetal deaths are associated with unnoticed infections and other diseases in the middle of pregnancy [7–10].

Antenatal care remains a significant public health intercession for preventing maternal and neonatal mortality worldwide, by enhancing the chance of accessing life-saving obstetric care [11, 12]. Timely and appropriate antenatal practices have a life-saving potential for mothers and children [13, 14]. The World Health Organization recommends that all pregnant women should receive at least four focused antenatal check-ups and start their primary visit during the first trimester with a skilled health worker. Moreover prescribes that to make the service a compelling preventive measure, the content, and quality of care should also be observed [13, 15–17].

Evidence has shown that ANC alone can reduce maternal mortality by 20%. Good quality regular attendance [18] and early initiation of ANC are associated with better maternal health outcomes [19, 20]. Adequate antenatal care services are underutilized in developing nations [18, 19]. Although most of the women (88%) had at least one visit, a small proportion of pregnant women (40%) in these nations attended the recommended four visits [20]. Furthermore, in sub-Saharan Africa, nearly three-quarters (72%) of women initiate their first ANC check-ups after the first trimester of pregnancy [21].

Previous studies in Ethiopia reported that the proportion of mothers who initiated their first antenatal visit in the first trimester ranged from 12% to -33% and less than 30% of mothers had four or more checkups throughout their pregnancy [22–25]. A similar finding was reported in a national survey [26] where only 32% of mothers had received four or more ANC check-ups.

Furthermore, local studies from Debretabor and Addis Ababa revealed that approximately 20.2% and 10.1% of women received sufficient content throughout pregnancy and 11% and 2.6% received overall adequate ANC services, respectively [23, 27]. To attain national goals or meet international standards, ANC services should be sufficient, in line with the recommendations of national protocols or WHO guidelines [27].

To the best of our knowledge, there is no literature available in the study area, and it was difficult to reach enough studies conducted in other areas of Ethiopia on the overall adequacy and determinants of ANC whose results are based on the combined indicators including the time of the first visit, the number of visits, and items of ANC services.

Unlike the majority of previous studies assessing utilization of ANC only from the perspective of the frequency of attendance, the current study provides a more comprehensive picture

of ANC adequacy and its relations to selected factors by indexing three indicators: the frequency of visits, the timing of the first visit, and service content utilization. An accurate assessment of ANC adequacy will contribute to the development of public health interventions aimed at improving ANC accessibility, thereby improving birth outcomes. It is predominantly vital if Ethiopia is to attain the Sustainable Development Goal 3 (SDG 3) to reduce the maternal mortality ratio (MMR) to less than 70 per 100,000 live births by 2030 [5, 6].

Empirical findings also indicate that socio-demographic, obstetric, and health system factors are associated with the utilization of ANC services [22–25]. However, there is a need to examine these factors systematically within one study using the Andersen and Newman Behavioral Model (ANBM), to identify the most important barriers and enablers for adequate ANC utilization [28, 39]. Moreover, the association between ANC service utilization and provider and patient approach was not addressed in a similar previous study in Ethiopia even though service content might be influenced by the service providers' approach; it is expected that a clear understanding of such factors is crucial in building a responsive maternal health care system to improve health outcomes in Ethiopia.

## Materials and methods

### Study design and setting

A community-based cross-sectional study design was employed from November 20[th] toDecember 31[st], 2019, in Wolaita Sodo town, 380 km to the South of Addis Ababa. The town has an estimated total population of 110,659; 48% were female according to population projections for Ethiopia published by the central statistical agency by 2014 [30]. The town has three sub-cities, each having approximately ten kebeles within it. It has one governmental referral hospital, one private hospital, and 21 private clinics, and comprises four health centers and 11 health posts for people residing in the town at different locations within the city. In each kebele, three to four urban Health Extension Workers (HEWs) provide health promotion activities.

### Population

The study population consisted of reproductive-aged women who gave birth within 1–2 years preceding the survey regardless of the birth outcome and had a minimum of one ANC follow-up for the pregnancy of the most recent baby. Women who had lived in the study area for over 6 months, and delivered their most recent baby after 28 weeks of gestation were included. Women, who were critically ill and mentally impaired during the information collection period, were excluded from the study.

### Measurement variables

**Outcome variable.** The outcome variable of this study was ANC utilization. Antenatal care utilization was constructed using three ANC utilization attributes: frequency of ANC visits, the timing of the primary visit, and the number of service contents that the mother received throughout the pregnancy of recent birth. The overall antenatal care utilization was considered adequate if the mother received all three attributes adequately otherwise inadequate.

The frequency of visits was considered adequate if the mother had a minimum of 4 visits during her pregnancy; and the timing of the primary visit was deemed adequate if the primary visit was within the first 12 weeks of pregnancy [13, 15]. Service content was assessed by asking participants about the elementary ANC components (weight and height, blood pressure, fundal or uterine height, and fetal heartbeat, urine and blood samples taken (blood type, hemoglobin (anemia) and syphilis test), tetanus injection, iron supplementation, and information or

counseling was given about signs of pregnancy complications) received during pregnancy of the most recent birth in their first ANC visit. Service content was considered adequate if the mother received more than or equal to the mean (8) of the 12 mentioned services otherwise inadequate.

**Predictor variables.** The predictor variables were conceptualized based on the Andersen Newman behavioral model of health care utilization [28, 29], and then clustered into three sets of factors: predisposing, enabling, and need factors as shown in [Table 1].

## Sample size and sampling procedure

To determine the sample size a single population proportion formula was used with the following assumptions; 15.3% [22] proportion of attending four or more visits, 95% confidence interval, 5% margin of error, and design effect of 2. For possible non-response, 10% was added and the final sample size was 670. The samples were allocated to all sub-cities proportional to the number of eligible women. Households of qualified women were first identified with the help of Health Extension Workers (HEWs) who kept a list, including contact details, of women. When two or more eligible women were found within the selected household, one was selected using the Kish grid selection procedure.

## Source of data and data collection methods

A structured questionnaire adapted from existing surveys [22, 27] and the Anderson and Newman Behavioral Model of health service utilization [29, 30] were used as a guide to avoiding missing potential independent variables. Before the administration of the questionnaire, it was reviewed by two senior experts and pilot tested with 35 women who had similar characteristics to the study population. Item questions were checked for reliability with a Cronbach's alpha of 0.976.

The questionnaire was translated into the local language (Amharic) and subsequently translated back to English by different language experts to check for internal consistency. A template of the study tool was prepared using an online survey tool (open data kit (ODK)) and downloaded onto tablets for offline data collection. Data collection was facilitated by 12 health extension workers (HEWs) and 3 supervisors with extensive experience in conducting interviews using electronic devices (tablets).

## Data quality control

The study instrument was pre-tested on 35 similar study populations in the town of Hosanna. All required revisions were made to the tool based on the pre-test results. Experienced enumerators: 12 HEWs, a BSc nurse, and 2 midwives were recruited for data collection and supervision. A two-day intensive training on the aim of the study and sampling procedures was provided to the enumerators. To reduce recall bias we used a recall period of up to two years similar to the United Nations Children's Fund supported Multiple Indicator Cluster Surveys (MICS) [31, 32] which is far shorter than that for Demographic and Health Surveys (up to five years). We used tablets for data collection to avoid missing or incomplete responses.

Selected women were oriented about the study and their random selection, and data from participants were received in private settings after deep discussion that removed their doubts and cleared their confusion. Supervisors cross-checked the completed responses on the tablet by repeating the interviews with 10% of the respondents to check for correct completion of responses. The responses were uploaded into the online survey tool daily and the principal investigator double-checked for any inconsistencies and gave feedback to the interviewers

**Table 1. Determinant variables for adequate antenatal care service and their operational definitions.**

| Variable category | Operational Definitions |
|---|---|
| **Predisposing factors** | |
| Maternal education | Formal schooling status of women starting from elementary school |
| Marital status | Marital status of the women at the time she got pregnant for her most recent birth |
| Maternal employment status | Women employment status while she got pregnant for her most recent birth |
| Partner education | formal schooling status starting from elementary school |
| Age at pregnancy | Age in completed years while the respondent got pregnant at the most recent birth |
| Mass media exposure | Respondents exposure to television, radio, and internet |
| Previous use of ANC | The practice of ANC for any of the previous pregnancies in the woman's life |
| Best friends use of maternal care | A friend who regularly shared the woman's feelings, emotions, and opinions and various behavioral practices important in the woman's life and uses services such as ANC, delivery care and PNC |
| Number of live births | Total number of live births in women's life |
| Number of pregnancy | Total number of pregnancies including stillbirth/abortion in women's life |
| **Enabling factors** | |
| Husband support | Respondent's judgment about husband's support that encourages her to follow ANC |
| Wealth index | Produced from the existing variables (household assets ownership, household characteristics, and access to utilities) from the data set through factor analysis using Principal Component Analysis (PCA) |
| involvement in decision making | Maternal involvement in the last to say yes/no for ANC |
| Type of health facility received ANC service | The health facilities where the mother received ANC for the pregnancy of most recent birth |
| **Need factor** | |
| Awareness about pregnancy-related complications | Respondents were asked a question that consisted of a multiple-choice response about pregnancy-related complications, and those correctly mentioned $\geq$ to 5 were categorized as having good knowledge about it. |
| Awarness about danger signs of pregnancy | Respondents were asked a question consisted multiple-choice responses about danger signs of pregnancy(such as bleeding) and those correctly mentioned $\geq$5 were categorized as having good knowledge about it. |
| Awareness about starting time of pregnancy | Respondents were asked about the recommended time to start ANC follow-up during pregnancy: Good awareness if they answered $\leq$12 weeks otherwise poor awareness |
| Awareness about the frequency of recommended ANC visit | Respondents were asked about the minimum number of recommended ANC visits throughout the pregnancy: Good awareness if answer the question correctly($\geq$4) |
| Perception of ANC importance | Respondents were asked their opinions about the importance of ANC attendance and those who respond "very important" were categorized as having good perception. Those who respond"somewhat important" and"not important" and those who said" I do not know" were categorized as having a poor perception of ANC. |
| Perception of provider-patient approach | Respondents were asked 9 questions that assess their perception of the 'provider-patient approach'. Every single question comprised 'agree','neutral', and 'Disagree' options, and those who answered agree scored 2, neutral scored 1, and disagree given score 0 for each question. Those who scored a total sum of >12, 7–12, and $\leq$ 6 categorized as having good perception, moderate perception, and bad perception, respectively |
| Pregnancy intention | Respondents were asked to whether their pregnancy of last birth was planned or not; labeled as "Intended" if the pregnancy was wanted at the time, "Unintended" If the pregnancy was wanted later /not wanted at all |
| History of a risky pregnancy | Lifetime experience of abortion/stillbirth and other pregnancy-related complications |

daily. Before commencing the data analysis, appropriate transformations were made on the variables. Missing values at random were also managed using multiple imputations.

## Data processing and analysis

The data were directly exported from the digital survey tool to STATA software version 15 for analysis. Descriptive statistics were also calculated. Both bivariable and multivariable logistic regression analyses were performed. In the bivariable analysis, simple summary statistics (percentage of the outcome variables were obtained for each category of the selected explanatory variables to examine the unadjusted but statistically significant relationship between dependent variables and selected explanatory variables. Statistical significance was tested using bivariable logistic regression.

Factors that were significant with a p-value of less than 0.05 in bivariable logistic regression were retained for further consideration with a multivariable logistic regression to control for confounders. Four different models were fitted to identify the factors determining the adequacy of ANC services: the timing of the first visit, number of visits, service content, and overall adequacy. The odds ratios and 95% confidence intervals were computed and a p-value of less than 0.05 was used to determine the cut-off points for statistical significance. The necessary assumption of model fitness during logistic regression was checked using the Hosmer and Lemeshow goodness of fit test statistics. Multicollinearity was checked by a variable inflation factor and all showed no multicollinearity with a variable inflation factor of less than five.

## Ethical approval and consent to participate

Ethical clearance was obtained from the academic research directorate of Wolaita Soddo University, college of health scienceand medicine, and the official letter was written to each sub-city. Written informed consent was also obtained from the study participants and parents of participants under the age of 16 before interviewing. No personal details were recorded or produced on any documentation related to the study and privacy was assured. No one was obliged to participate unless otherwise agreed to take part.

## Results

### Socio-demographic and reproductive characteristics of the participants

A total of 670 respondents aged between 13 and 41 years participated yielding a response rate of 100%. The majority of the women 561(83.73%) were under the age of 35 at the time of pregnancy of the last child and 64.9% of participants fell within the age group of 25–34 years. Most mothers attended at least primary education 554 (82.69%), not employed 536 (80%), and married 600 (89.55%), and had a husband who attended at least primary education 547 (91.17%). More than half of the mothers had unemployed husbands 335(59.17%), had no mass media exposure 401(59.85%), and decided on maternal health jointly with their husbands 431 (64.33%). About 223(33.28%) mothers belonged to the high wealth index group [Table 2].

A total of, 427(63.73%) participant's pregnancy of the most recent birth was wanted whereas, 536(80.00%) participants claimed that they had a history of previous ANC follow-up. Of all the study participants, 476(71.04%), 547(81.64%), and 430(64.18%) participants revealed that they received ANC from the public health facilities, received ANC from the skilled health providers, and their major source of information about ANC were health professionals, respectively. About 263(39.25%) and 287(42.84%) mothers disclosed that they experienced a riskypregnancy and had best friends who utilized maternity care, respectively [Table 2].

**Table 2. Socio-economic and reproductive characteristics of women in Southern Ethiopia, 2019.**

| Variables | Category | Total(N) and (%) |
|---|---|---|
| Age at birth of last-child (N = 670) | 15–24 | 126(18.80%) |
| | 25–34 | 435(64.93%) |
| | 35–49 | 109(16.27%) |
| Educational status (N = 670) | No formal education | 116(17.31%) |
| | Primary education | 190(28.40%) |
| | Secondary education | 193(28.81%) |
| | Tertiary and above | 171(25.52%) |
| Occupational status (N = 670) | Employed | 134(20.00%) |
| | Unemployed | 536(80.00%) |
| Religion (N = 670) | Protestant | 358(53.43%) |
| | Orthodox | 214(31.94%) |
| | Adventist | 69(10.30%) |
| | Catholic | 17(2.54%) |
| | Muslim | 12(1.79%) |
| Marital status (N = 670) | Married | 600(89.55%) |
| | Single | 70(10.45%) |
| Partner's educational status (N = 600) | No formal education | 53 (8.83%) |
| | Primary education | 79 (13.17%) |
| | Secondary education | 260(43.33%) |
| | Tertiary and above | 208(34.67%) |
| Partner's occupational status (N = 600) | Employed | 245(40.83%) |
| | Unemployed | 355(59.17%) |
| Wealth index (N = 670) | Lower | 221(32.99%) |
| | Medium | 226(33.73%) |
| | Higher | 223(33.28%) |
| Exposure to mass media (N = 670) | Almost daily exposure | 269(40.15%) |
| | Infrequent/not at all | 401(59.85%) |
| Involvement in decision making (N = 670) | Not involved at all | 13(1.94%) |
| | Partially involved | 431(64.33%) |
| | Fully involved | 226(33.73%) |
| Pregnancy intension (N = 670) | Unintended | 243(36.3%) |
| | Intended | 427(63.7%) |
| History of previous ANC follow up (N = 670) | No | 197(29.3%) |
| | Yes | 473(70.7%) |
| Having husband support in using maternal health care service (N = 600) | No | 85(14.17%) |
| | Yes | 515(85.83%) |
| Having a best friend who uses maternal health care service | No | 383(57.16%) |
| | Yes | 287(42.84%) |
| Site of ANC received for the most recent birth | Governmental | 476(71.04%) |
| | Private | 194(28.96%) |
| Source of information about ANC | Mass media | 49(7.3%) |
| | Friend/relative/ neighbor | 186(27.8%) |
| | Health professionals | 435(64.9%) |
| Health provider | Unskilled | 123(18.4%) |
| | Skilled | 547(81.6%) |

(*Continued*)

**Table 2.** (Continued)

| Variables | Category | Total(N) and (%) |
|---|---|---|
| Number of live children | 1 | 129(19.25%) |
| | 2–3 | 418(62.39%) |
| | >3 | 123(18.36%) |
| Parity | 1 | 99(14.78%) |
| | 2–3 | 363(54.18%) |
| | >3 | 208(31.04%) |
| History of a risky pregnancy | Yes | 263(39.25%) |
| | No | 407(60.7%) |
| Availability of health worker at home | Yes | 540(80.6%) |
| | No | 130(19.4%) |

## Knowledge and perception of participants

Of all the study participants, 269(40.15%), and 419(62.54%) participants had good knowledge of the timing of the first ANC visit and the number of recommended visits throughout the pregnancy. The majority of mothers had a good perception of ANC importance 575(85.82%). Only 34.33% and 40.60% of the participants had good knowledge about pregnancy-related complications and danger signs, respectively. About 30.45% and 34.48% of mothers disclosed that the provider-patient approach during ANV visits was medium and very good, respectively. Only 140(20.90%) mothers had a good perception of patient waiting time at health institutions to receive ANC services [Table 3].

**Table 3. Knowledge and perception of women about ANC service in Southern Ethiopia, 2019.**

| Variable | Category | Total (N) and % |
|---|---|---|
| Awareness about the timing of first visit | Good awareness | 269 (40.15%) |
| | Poor awareness | 401 (59.85%) |
| Awareness about the frequency of visits | Good awareness | 419 (62.54% |
| | Poor awareness | 251 (37.46%) |
| Awareness about pregnancy-related complications | Good awareness | 230 (34.33%) |
| | Poor awaremess | 440 (65.67%) |
| Awareness about danger signs of pregnancy | Good awareness | 272 (40.60%) |
| | Poor Awareness | 398 (59.40%) |
| The main source of information about ANC | Mass media | 54 (8.06%) |
| | Family/friends/relatives | 186 (27.76%) |
| | Health professionals | 430 (64.18%) |
| Perception towards ANC importance | Very important | 575 (85.82%) |
| | Somehow important/not important | 95 (14.18%) |
| Perception towards the provider-patient approach | Good perception | 231 (34.48%) |
| | Moderate perception | 204 (30.45%) |
| | Bad perception | 235 (35.07%) |
| Perception towards waiting time | Long | 185 (27.61%) |
| | Moderate | 345 (51.49%) |
| | Short | 140 (20.90%) |
| Perception about cost | Expensive | 87(12.99%) |
| | Inexpensive | 583(87.01%) |

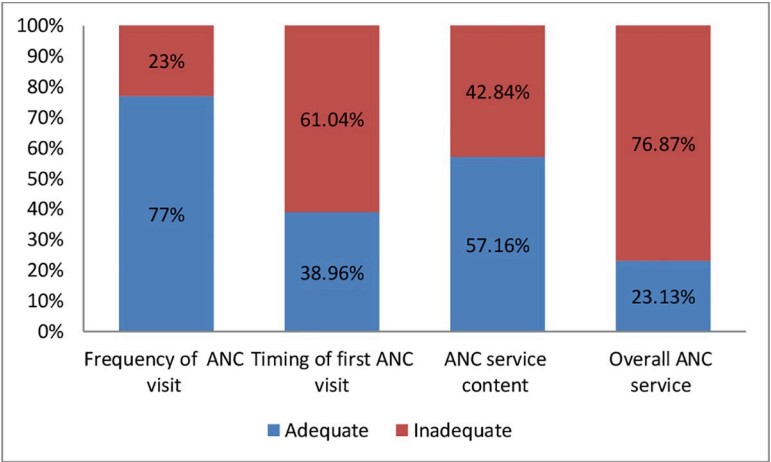

**Fig 1. Distribution of respondents according to antenatal utilization attributes among mothers in Southern Ethiopia, 2019.**

## Levels of adequate ANC utilization

While we observed the distribution of participants according to ANC utilization, only 38.96% of participants initiated antenatal care service early, whereas 77.01% of participants had more than four ANC visits throughout the pregnancy of the most recent birth. Three hundred eighty-three (57.16%) participants received adequate service items and only 23.13% utilized overall antenatal care services adequately [Fig 1].

Regarding individual ANC components, weight, blood pressure, and fetal heart rate measurement were the most common items received by 95.82%, 95.22%, and 90.15% of respondents, respectively. The majority of participants claimed that they had urine exam tests (82.69%), blood type tests (74.78%), and received iron supplementation (72.24%). Slightly more than half of the participants reported that they had received the tetanus toxoid vaccine (51.94%) and symphysis-fundal measurements (50.60%). Less than half of the participants disclosed that they had an anemia test (42.54%) and a syphilis test (41.04%). The least received ANC components as reported by the participants were height measurement (23.73%), and counseling about pregnancy-related complications (28.51%) [Fig 2].

## Variations in the adequacy of ANC service

Table 3 presents the percentage distributions of the adequacy of ANC services, the timing of the first visit, the frequency of visits, and the adequacy of service items across a set of selected explanatory variables. Variables that showed a statistically significant difference in the bivariable model are indicated by asterisks. The findings revealed that the difference in ANC coverage across the categories of maternal education did not show consistent variation.

Women who attended secondary and ≥ tertiary education were more likely to attend ANC early (44.04%), (61.40%), to have an adequate number of visits (82.38%), (88.89%), to receive acceptable service contents (67.36%), (71.93%) and to have overall adequate ANC (24.35%) 45.03%), respectively compared to (33.62%) attended ANC early, (68.96%) had an adequate number of visits, (45.69%) received acceptable service contents, and (12.93%) had overall adequate ANC for those who never attended any formal education. However, those who attended primary education were less likely to attend antenatal care early (16.84%), to have an adequate number of visits (66.32%), to have sufficient service content

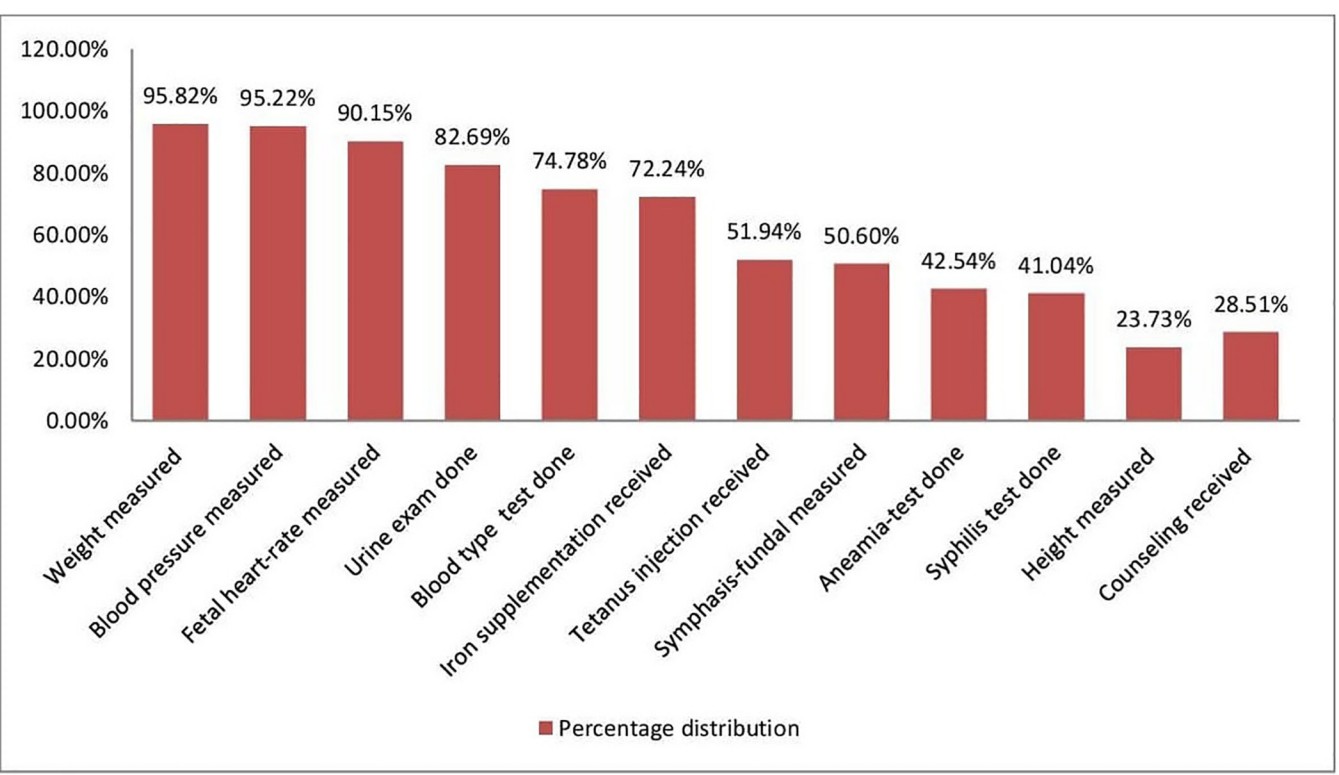

**Fig 2. Percentage distribution of recommended ANC service contents received by mothers in southern Ethiopia, 2019.**

(40.53%), and to have adequate ANC services (8.42%) compared to those who never attended any formal education.

Of all the participants who had an intended pregnancy at the last recent birth, 50.00%, 88.03%, 67.37%, and 31.46% started ANC early, had an adequate number of visits, received sufficient service content, and had overall adequate ANC services compared to 19.44%, 58.20%, 39.34%, and 8.61, respectively of those who had an unintended pregnancy. women whose best friend used maternal care also had a higher chance of starting ANC early (50.52%), having an adequate number of visits (91.39%), receiving sufficient service contents (79.44%), and having overall adequate ANC services (39.37%) than 30.29%, 66.58%, 40.47%, and 10.97%, respectively of those whose best friend did not utilize maternal care.

Women whose husbands/partners supported them to use maternal care were more likely to attend ANC early (43.88%), to have an adequate number of visits (81.75%), to receive sufficient service contents (63.30%), and to have the overall adequate ANC services (26.80%) compared to 10.59%, 41.18%, 20.00%, and 3.53%, respectively of women who did not support by their husband.

Those who had a history of risky pregnancy, and those who received ANC from skilled health providers had higher chances of starting ANC early (54.37%), (42.49%), having an adequate number of visits (88.59%), (79.49%), receiving sufficient service contents (74.90%), (59.89%), and having overall adequate ANC services (36.12%), (26.19%), respectively compared to those who had not a history of risky pregnancy and those who received ANC from the unskilled providers started ANC early (29.00%), (23.39%), have an adequate number of visits (69.78%), (66.94%), received sufficient service contents (45.70%), (45.16%), and have overall adequate ANC services (14.74%), (9.68%), respectively.

Moreover, women who received ANC from the private health facilities and women who belonged to a high wealth index group had higher chances of starting ANC early (52.58%), (55.61%), having an adequate number of visits (79.38%), (82.06%), receiving sufficient service contents (64.95%), (76.23%), and having overall adequate ANC services (41.24%), (45.29%), respectively than those who received ANC from private health facilities and those belonged to low wealth index group started ANC early (33.40%), (26.70%), have an adequate number of visits (76.26%), (66.06%), have the sufficient service contents (54.00%), (35.75%), and have the overall adequate ANC services (15.76%), (9.50%), respectively.

Women, who had almost daily exposure to mass media and had a good awareness about pregnancy-related complications were more likely to initiate ANC early, to have an adequate number of visits, and to receive acceptable service content throughout the pregnancy. The number of alive children and pregnancy showed a positive association with the sufficient number of service items. Women who had three or more, alive children and lifetime pregnancy were more likely to have sufficient service content. Of the participants who had a previous history of ANC follow-up, 59.81% of participants received a sufficient number of service contents compared to 46.67% of those who had not a previous history of ANC follow-up [Table 4].

## Determinants of adequate ANC utilization

Table 5 showed factors associated with the timing of the first visit, the frequency of visit, ANC service content, and overall adequacy of ANC. This table presented only variables that remain significant after controlling for potential confounders in the multivariable model. Maternal education was the predisposing factor that showed a significant association with the timing of the first visit, ANC service content, and overall adequacy of ANC service. Mothers who attended secondary education were 1.96 times more likely to receive acceptable number service contents (AOR,1.96;95%CI: 1.07, 3.62) than those who never attended any formal education. Mothers who attended tertiary and above education were 2.33, 2.03, and 4.15 times more likely to initiate ANC early, to receive sufficient service content, and to have overall adequate ANC service (AOR,2.33;95%CI: 1.29, 4.19), (AOR,2.03;95%CI: 1.11, 3.72), and (AOR,4.15,95% CI: 1.95, 8.83), respectively compared to those who never attended formal education.

Having a best friend who used maternal care was another predisposing factor significantly associated with the frequency of visits, ANC service contents, and overall adequacy of ANC service. Women whose best friend utilized maternal care were 3.31, 3.20, and 2.01 times more likely to have the adequate number of visits, sufficient service content, and overall adequate ANC (AOR, 3.31;95% CI: 1.89, 5.81), (AOR,3.20;95%CI: 2.03, 5.03), and (AOR,2.01;95% CI: 1.18, 3.41) than those whose best friends did not utilize maternal care.

Enabling factors such as husband support, type of health facilities received ANC, and wealth index were significant predictors of ANC service utilization. Women whose husbands/partners supported them to follow ANC were 2.89, 3.42, 4.62, and 3.84 times more likely to start ANC visit early, to have an adequate number of visits, to have an adequate service content, and to have overall adequate ANC service (AOR, 2.89; 95% CI: 1.31, 6.37), (AOR, 3.42; 95% CI: 1.89, 6.20), (AOR,4.62;95%CI: 2.40, 8.91), and (AOR,3.84; 95%CI: 1.05, 14.08) than their counterparts.

Women who had ANC follow-ups in private health facilities were more likely to have overall adequate ANC services (AOR, 2.27;95% CI:1.33, 3.88) than those who had a follow-up in public health facilities. Moreover, women who belonged to high wealth index were 1.82, 3.67, and 3.61 times more likely to start ANC early, to receive sufficient service content, and to have overall adequate ANC (AOR,1.82;95%CI: 1.06, 3.13), (AOR,3.67;95%CI: 2.13, 6.31), and (AOR,3.61; 95%CI: 1.86, 6.99) than those who belonged to low wealth index group.

**Table 4. Percentage distribution of ANC service utilization with selected variables in Southern Ethiopia, 2019.**

| Variables | Category | Early initiation | Adequate number of visit | Adequate service content | Overall adequate |
|---|---|---|---|---|---|
| Educational status of the mother | No education | 33.62*** | 68.96*** | 45.69*** | 12.93*** |
| | Primary education | 16.84 | 66.32 | 40.53 | 8.42 |
| | Secondary education | 44.04 | 82.38 | 67.36 | 24.35 |
| | Tertiary education | 61.40 | 88.89 | 71.93 | 45.03 |
| Occupational status of the mother | Employed | 58.96*** | 87.31*** | 70.90*** | 41.04*** |
| | Unemployed | 33.96 | 74.63 | 53.73 | 18.66 |
| Marital status | Married | 39.17 | 76.00* | 57.17 | 23.50 |
| | Single | 37.14 | 87.14 | 57.14 | 20.00 |
| Occupational status of husband/partner | Employed | 46.94** | 87.31 | 70.90 | 30.20** |
| | Unemployed | 33.80 | 74.62 | 53.73 | 18.87 |
| Number of alive children | 1 | 37.21 | 70.54 | 50.39* | 25.58 |
| | 2–3 | 37.41 | 79.86 | 56.59 | 19.42 |
| | >3 | 45.97 | 75.00 | 66.13 | 33.06 |
| Parity | 1 | 40.40 | 75.76 | 52.53** | 27.27 |
| | 2–3 | 37.47 | 79.61 | 53.17 | 20.94 |
| | >3 | 40.87 | 73.56 | 66.35 | 25.00 |
| Exposure to mass media | Almost daily | 47.58*** | 84.15*** | 55.02 | 21.56 |
| | No exposure | 33.17 | 72.57 | 58.60 | 24.19 |
| Pregnancy intension | Intended | 50.00*** | 88.03*** | 67.37*** | 31.46*** |
| | unintended | 19.67 | 58.20 | 39.34 | 8.61 |
| Having a best friend who used maternal care | Yes | 50.52*** | 91.39*** | 79.44*** | 39.37*** |
| | No | 30.29 | 66.58 | 40.47 | 10.97 |
| Previous history of ANC follow-up | Yes | 40.00 | 77.94 | 59.81** | 22.80 |
| | No | 34.81 | 74.07 | 46.67 | 24.44 |
| Having husband support | Yes | 43.88*** | 81.75*** | 63.30*** | 26.80*** |
| | no | 10.59 | 41.18 | 20.00 | 3.53 |
| Having a history of risky pregnancy | Yes | 54.37*** | 88.59*** | 74.90*** | 36.12*** |
| | no | 29.00 | 69.78 | 45.70 | 14.74 |
| Awareness about pregnancy-related complications | Good awareness | 50.00*** | 87.40*** | 62.61* | 23.04 |
| | Poor awareness | 33.18 | 71.81 | 54.32 | 23.18 |
| Awareness about trimester of the first visit | Good awareness | 48.70*** | 78.81 | 63.20* | 30.86*** |
| | Poor awareness | 32.42 | 76.06 | 53.12 | 18.00 |
| Awareness about frequency of visit | Good awareness | 44.15*** | 83.53*** | 58.00 | 24.11 |
| | Poor awareness | 30.28 | 66.53 | 55.78 | 21.51 |
| Type of health facilities received ANC | Private | 52.58*** | 79.38 | 64.95** | 41.24*** |
| | public | 33.40 | 76.26 | 54.00 | 15.76 |
| Health provider | Skilled | 42.49*** | 79.49** | 59.89** | 26.19*** |
| | Unskilled | 23.39 | 66.94 | 45.16 | 9.68 |
| Perception of provider–patient approach | Good | 48.92** | 80.52 | 60.61* | 31.17** |
| | Moderate | 34.31 | 77.94 | 60.29 | 16.18 |
| | Not good | 33.19 | 73.19 | 51.06 | 21.28 |
| Wealth index | High | 55.61*** | 82.06*** | 76.23*** | 45.29*** |
| | Medium | 34.51 | 83.19 | 59.29 | 14.60 |
| | Low | 26.70 | 66.06 | 35.75 | 9.50 |

**Table 5. Results of adequate ANC service utilization to identify determinants of adequate ANC service utilization in Southern Ethiopia, 2019.**

| Variables | Indicators of ANC | | | |
|---|---|---|---|---|
| | Early initiation | Adequate number of visit | Adequate service content | Overall adequate ANC |
| Maternal education(RC = No education) | | | | |
| Primary education | 0.54 (0.28, 1.02) | | 1.19(0.66, 2.16) | 0.81(0.33, 1.96) |
| Secondary education | 1.59 (0.88, 2.89) | | 1.96(1.07, 3.62)* | 2.01(0.92, 4.38) |
| Teritiary and above | 2.33(1.29, 4.19)** | | 2.03(1.11, 3.72)* | 4.15(1.95, 8.83)*** |
| Pregnancy intension (RC = Unintended) | | | | |
| Intended | 2.79 (1.72, 4.54)*** | 4.91(3.00,8.02)*** | 1.96(1.25, 3.07)** | 2.60(1.35, 4.99)** |
| Having a best friend who uses maternal care(RC = No) | | | | |
| Yes | | 3.31(1.89, 5.81)*** | 3.20(2.03, 5.03)*** | 2.01(1.18, 3.41)* |
| Husband support(RC = No) | | | | |
| Yes | 2.89(1.31, 6.37)** | 3.42(1.89, 6.20)*** | 4.62(2.40, 8.91)*** | 3.84(1.05, 14.08)* |
| Having a history of risky pregnancy(RC = No) | | | | |
| Yes | 2.49(1.63, 3.81)*** | 1.94(1.14,3.30)* | | 2.59(1.55, 4.35)*** |
| Awareness of the frequency of visit (RC = poor awareness) | | | | |
| Good awareness | | 2.51(1.52,4.13)*** | | |
| Skilled Service Provider | | 0.52(0.28,0.95)* | | |
| Site of health facilities recived ANC(RC = Public) | | | | |
| Private | | | | 2.27(1.33, 3.88)** |
| Wealth index (RC = Low | | | | |
| Medium | 0.79(0.47,1.34) | | 1.28(0.78, 2.12) | 0.91(0.45,1.82) |
| High | 1.82(1.06, 3.13)* | | 3.67(2.13, 6.31)*** | 3.61(1.86, 6.99)*** |

Key: RC:Reference Category.

*: statistically significant variables with p-value <0.05

**: statistically significant variables with p-value <0.01

***: statistically significant variables with p-value <0.001

Moreover, having a history of risky pregnancy, having awareness about the frequency of recommended ANC visits, and having pregnancy intension were among the need factors that significantly predict ANC service utilization. Women who had a history of risky pregnancy were 2.49, 1.94, and 2.59 times more likely to start ANC early, to have a sufficient number of visits, and to have overall adequate ANC services (AOR,2.49;95% CI: 1.63, 3.81), (AOR,1.94;95%CI: 1.14,3.30), and (AOR,2.59; 95%CI: 1.55, 4.35) than those with no history of a risky pregnancy.

Women who had a good awareness about the recommended number of ANC visits had more odds of attending four or more visits (AOR,2.51;95% CI:1.52, 4.13) than their counterparts. Women who had a planned pregnancy of most recent birth were 2.79, 4.91, 1.96, and 2.60 times more likely to start ANC early, to have an adequate number of visits, to have sufficient service contents, and to have overall adequate ANC services (AOR, 2.79; 95% CI: 1.72, 4.54), (AOR,4.91; 95% CI: 3.00, 8.02), (AOR,1.96; 95% CI: 1.25, 3.07), and (AOR,2.60;95% CI: 1.35, 4.99) compared to women who had unplanned pregnancy [Table 5].

## Discussion

This study assessed the magnitude and determinants of adequate antenatal care service utilization among the mothers in southern Ethiopia using the three ANC service adequacy indicators i.e. the timing of the first visit, the frequency of visit, and ANC service items. The number and the timing of antenatal visits and the prescribed service items received amid antenatal visits are

very important in recognizing pregnancy risks conjointly with the management of delivery complications [33]. However, there is not enough literature available in Ethiopia, which assesses the magnitude and determinants of adequate ANC service utilization based on the combined three ANC service adequacy indicators.

Early initiation of ANC enables pregnant women to have an adequate number of visits and sufficient ANC service items [33]. In this study, of women who initiated their first ANC early, 63.22% received sufficient service items, and 93.10% had an adequate number of visits compared to 53.30% and 67.00%, respectively of those who had late ANC booking. However, only 38.96% of women initiated antenatal visits within the 12th week of gestation. The proportion of women who had early ANC visits in this study is lower than in a study conducted in Addis Ababa where 50.3% of women started ANC early [27]. This indicated that despite the government is dedicated to increasing the coverage, the focus was not given to the starting time of ANC. So, raising the ANC coverage only may not be satisfactory to pregnant women rather they should utilize the service at the recommended time of gestational age.

This study revealed that the ANC utilization rate in terms of having an adequate number of ANC visits was much higher than EDHS 2016 [26]. The exhibited difference in the magnitude of an adequate number of ANC visits might be because of the differences in study time which was three years later than the EDHS report.

The proportion of women who attended more than four visits in the current study was also higher than the findings from many other previous studies conducted in different regions of Ethiopia [22, 23, 25, 27]. These disparities might be attributed to the fact that the studies were conducted at different points in time and the level of safe motherhood activities might be different in different parts of Ethiopia.

Furthermore, 57.16% and 23.13% of women in this study received sufficient service content and had overall adequate ANC service, respectively. This was higher than previous studies conducted in Ethiopia [23, 27]. The magnitude of the overall adequate ANC service was also higher than the study done in Nigeria where only 15.5% of mothers who utilized a skilled provider received the three ANC attributes adequately [34].

The exhibited disparities in the magnitude of ANC utilization among these studies might be due to differences in study time and population, where there are visible variations in socio-cultural background and destitute community discernments towards utilization of ANC that would likely have decreased ANC uptake. Moreover, health professional's commitment towards ANC service arrangement has considerable significance for the variation in these studies.

Educational status encompasses a significant positive association with utilization of ANC [23, 27, 35]. Consistent with these studies, the current study uncovered that women who attended secondary and tertiary education were 1.96 and 2.03 times more likely to have sufficient service content compared to those who never attended any formal education. Moreover, women who attended tertiary and above education were 2.33 and 4.15 times more likely to initiate ANC early and to have overall adequate ANC services, respectively. This might be because education is an imperative instrument that increases women's cognitive capacity which in turn increases knowledge of ANC, knowledge about pregnancy-related health issues, and increases their empowerment. Besides education helps women to transform their social value which empowers them to get access to maternal care through the reading of information and so on [35].

Women whose best friend utilized maternal care were 3.31, 3.20, and 2.01 times more likely to have the adequate number of visits, sufficient service content, and overall adequate ANC services, respectively than women whose best friend did not utilize maternal care. This has been illustrated in previous studies that acquiring and following advice from companions or

peers have facilitated better ANC utilization for pregnant women [22, 36]. Social systems that people develop, with their families, companions, or peer groups altogether impact the decision-making and health-seeking behavior or activities of pregnant women towards ANC, which eventually facilitates or delays their access to utilization of the service [28].

The finding of this study illustrated that the positive impact of best friends highlights a largely untouched asset for public health messaging in Ethiopia. Maternal health advancement exercises ought to include conveying messages concerning the advantage of attending ANC for the wellbeing of the mother and the child by focusing on the social networks of women through peer education programs.

Women whose husbands/partners supported them to follow ANC were 2.89, 3.42, 4.62, and 3.84, times more likely to start ANC visit early, to have an adequate number of visits, to have sufficient service contents, and to have overall adequate ANC services, respectively than their counterparts. This finding is in line with previous studies done in Ethiopia [22, 25]. This can be clarified by most women in the developing nations require a husband's endorsement to seek health services, including ANC; the spouse ought to have a steady demeanor towards health services, in this manner permitting his spouse to go to the services. Hence, intervention endeavors should be in place to improve the husband's support towards ANC to extend the uptake of the service.

Women received more adequate services and were more pleased with the services they gained from private facilities than public ones [27, 37]. In line with these studies, the current study revealed that women who received ANC from private health facilities were more likely to receive adequate ANC services than their counterparts. This might be since more educated and wealthier women in Ethiopia attended ANC in private health facilities.

Women who belonged to the high wealth index group were 1.82, 3.67, and 3.61 times more likely to start ANC early, to receive sufficient service contents, and to have overall adequate ANC than those who belonged to the low wealth index group. Local and foreign studies have found the same result [22, 34, 38]. Financial disparities in skilled ANC services remain the main challenge in Ethiopia [12], with the services being mainly used by the wealthier group. This can be rationalized with the prove that financial components seem to affect the health-seeking activities of women in many ways.

Women's experience of a risky pregnancy was a significant factor associated with adequate ANC utilization [23]. In this study, women who had a history of risky pregnancy were 2.49, 1.94, and 2.59 times more likely to start ANC early, to have a sufficient number of visits, and to have overall adequate ANC services than those who had no history of a risky pregnancy. The observed relationship might be since those who have an encounter of risky pregnancy know its consequences, in turn, tend to utilize ANC to avoid future abortion, stillbirth, and other pregnancy-related complications.

Moreover, the women whose recent birth was the result of planned pregnancy were 2.79, 4.91, 1.96, and 2.60 times more likely to start ANC early, to have an adequate number of visits, to have sufficient service contents, and to have the overall adequate ANC services compared to women with an unplanned pregnancy. The finding of this study is in line with previous studies [23, 25, 27, 38]. Unintended pregnancies and births have been appeared to claim great results to the mother and family and are global social and health burdens counting the US [39]. Mothers who planned their pregnancy are very cautious with their fetus's wellbeing and do their maximum endeavor and also talk with significant others to receive adequate ANC. Furthermore, women who had good awareness about the recommended frequency of ANC visits were more odds of attending four or more visits.

## Strengths and limitations of the study

Limiting the time frame for the inclusion of pregnancy occurred to two years preceding the survey could minimize the occurrence of recall bias. Assessing ANC utilization by considering the three essential attributes of ANC unlike the majority of previous studies that evaluated the utilization of ANC only from the perspective of the frequency of attendance was another strength of this study. This study considers many predictor variables from the three domains of ANBM for better control of confounders. On the other hand, service content was assessed by asking respondents about the contents they received during pregnancy of most recent birth but women may not specifically know the procedures, examinations, and laboratory investigations done for them despite efforts were made to minimize recall bias. Though training was given to the data collectors to inform respondents about the purpose of the study, the study might still be liable to social desirability bias since the data collection method was a face-to-face interview.

## Clinical and public health implication

Using all the three ANC service indicators to accurately reflect the overall adequacy of ANC utilization by women is preferable to using a single indicator. Many essential services that promote the health of mother and babies are mainly missed even in women who had an adequate number of visits [40]. To achieve the full life-saving potential that ANC promises for women and babies, adequate ANC services with a full package that provides essential evidence-based interventions are required.

The evidence from this study suggested that public health policies aimed at increasing adequacy of ANC utilization better to target risk groups discovered in our study such as; women with unintended pregnancies, non-educated, and economically-disadvantaged. Maternal health care providers should be influenced to provide ANC services with a full package without exempting those who had no history of a risky pregnancy because all women are potentially at risk for developing complications and should receive essential evidence-based interventions throughout their pregnancies.

The current study documented the issue of failure to utilize adequate ANC also attributed to where the service was provided. Majority of ANC attendees (81.6%) in this study visit public health facilities. Public health facilities are the main providers for the general population specifically for those economically disadvantaged and non-educated women groups. Strengthening maternal health care provider's adherence to ANC component guidelines and improving the quality of ANC at pubic health facilities is required by placing regular monitoring mechanisms to make sure that the recommendations of WHO for adequate ANC are met. The results have also highlighted that to improve ANC adequacy in the study area the expansion of services to include spouses has contributed significantly to the success of ANC utilization. Because in most areas of Ethiopia, there are challenges in increasing such health care service utilization mainly because the decisions that lead women to use the services seem to occur within the context of their marriage.

## Conclusion

The findings in the current study revealed that the magnitude of overall adequate antenatal care utilization was quite low. The level of adequacy of antenatal care services however is concealed by the high coverage in the number of antenatal visits. Predisposing factors (Maternal education and having a best friend who uses maternal care), enabling factors (husband support, site of health facility received ANC and wealth index), and need factors (pregnancy

intension and a risky pregnancy) played a key role in the adequacy of antenatal care service received by the women.

## Supporting information

**S1 File. Data collection tool for ANC section (English).**
(DOCX)

**S2 File. Data collection tool for ANC section (local language).**
(DOCX)

**S1 Data. Stata data result after multiple imputation.**
(DTA)

**S2 Data. Stata data without imputation.**
(DTA)

## Acknowledgments

We would like to express our deepest heartfelt thanks to Wolaita Soddo University for allowing us to conduct this study. Our special thanks go to the study participants for their commitment to give valuable information.

## Author Contributions

**Conceptualization:** Lielt Gebreselassie Gebrekirstos.

**Data curation:** Lielt Gebreselassie Gebrekirstos, Eyasu Alem Lake.

**Formal analysis:** Lielt Gebreselassie Gebrekirstos, Tsiyon Birhanu Wube.

**Funding acquisition:** Lielt Gebreselassie Gebrekirstos, Meron Hadis Gebremedhin.

**Investigation:** Lielt Gebreselassie Gebrekirstos, Tsiyon Birhanu Wube.

**Methodology:** Lielt Gebreselassie Gebrekirstos, Tsiyon Birhanu Wube, Eyasu Alem Lake.

**Project administration:** Lielt Gebreselassie Gebrekirstos, Tsiyon Birhanu Wube.

**Resources:** Lielt Gebreselassie Gebrekirstos, Tsiyon Birhanu Wube, Meron Hadis Gebremedhin, Eyasu Alem Lake.

**Software:** Lielt Gebreselassie Gebrekirstos, Tsiyon Birhanu Wube.

**Supervision:** Lielt Gebreselassie Gebrekirstos, Tsiyon Birhanu Wube, Meron Hadis Gebremedhin, Eyasu Alem Lake.

**Validation:** Lielt Gebreselassie Gebrekirstos, Eyasu Alem Lake.

**Visualization:** Lielt Gebreselassie Gebrekirstos, Tsiyon Birhanu Wube, Eyasu Alem Lake.

**Writing – original draft:** Lielt Gebreselassie Gebrekirstos.

**Writing – review & editing:** Lielt Gebreselassie Gebrekirstos, Tsiyon Birhanu Wube, Meron Hadis Gebremedhin, Eyasu Alem Lake.

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
