## [Decision Letter · Decision Letter 0]

15 Jul 2020

PONE-D-20-07041

PREDICTORS OF ADEQUATE ANTENATAL CARE SERVICE UTILIZATION AMONG MOTHERS IN SOUTHERN ETHIOPIA

PLOS ONE

Dear Dr. GEBREKIRSTOS,

Thank you for submitting your manuscript to PLOS ONE. After careful consideration, we feel that it has merit but does not fully meet PLOS ONE’s publication criteria as it currently stands. Therefore, we invite you to submit a revised version of the manuscript that addresses the points raised during the review process.

Your manuscript has been reviewed by three reviewers, who raise a number of concerns that must be addressed. Key among these is revising the Introduction to place this study in the appropriate context, discussing how the questionnaire was validated, and having your manuscript copy edited for language usage and grammar.

We look forward to receiving your revised manuscript.

Kind regards,

Emily Chenette

Staff Editor

PLOS ONE

Journal Requirements:

2. Please include additional information regarding the questionnaire used in the study and ensure that you have provided sufficient details that others could replicate the analyses. For instance, if you developed a questionnaire as part of this study and it is not under a copyright more restrictive than CC-BY, please include a copy, in both the original language and English, as Supporting Information.

3. In the Methods, please describe how the questionnaire was validated. If this did not occur, please provide the rationale for not validating the questionnaire.

5.Thank you for stating the following in the Funding Section of your manuscript:

[Wolaita Sodo University has covered the per diem for data collectors]

 [The author(s) received no specific funding for this work]

6.

We suggest you thoroughly copyedit your manuscript for language usage, spelling, and grammar. If you do not know anyone who can help you do this, you may wish to consider employing a professional scientific editing service.  

Reviewers' comments:

Reviewer's Responses to Questions

**Comments to the Author**

1. Is the manuscript technically sound, and do the data support the conclusions?

Reviewer #1: Partly

Reviewer #2: No

Reviewer #3: Partly

2. Has the statistical analysis been performed appropriately and rigorously? 

Reviewer #1: No

Reviewer #2: No

Reviewer #3: No

3. Have the authors made all data underlying the findings in their manuscript fully available?

Reviewer #1: Yes

Reviewer #2: No

Reviewer #3: No

4. Is the manuscript presented in an intelligible fashion and written in standard English?

Reviewer #1: No

Reviewer #2: No

Reviewer #3: No

5. Review Comments to the Author

Reviewer #1: Topic: Predictors of adequate antenatal care service utilization among mothers in Southern Ethiopia

Version 1

General comments

1. I advise to change the topics to Magnitude and determinants of adequate antenatal care utilization among mothers in Southern Ethiopia.

2. Abstract: better to report factor associated with adequate antenatal care utilization with OR with 95%CI

3. Introduction: The introduction to the study appears good. Authors should briefly explain what has been done so far. To be more interesting, author need to consider “what is the additional knowledge has is this study going to generate”? Try to explain the gap of the topic and the reason why you performed this research.

4. method: Concepts are well explained, but the following issues need to be addressed

• Your tool is adopted from the Anderson Newman behavioral model of health service utilization with some modification, so it needs tool validation, How to validate your tool?

• in the result section you present data related to knowledge and perception, how to measure knowledge and perception

• What are unique variables/factors/ examined compared to the pervious available studies. All identified variables are already addressed previously studies. therefore, include variable should not be redundant

• The authors do not explain how they checked for multicollinearity

• if the author plan to take p-value 0.2, as a cut of point for multivariable analysis, the author must present the result of identified variable with their p-value at bivariate model

5. Results: A result needs to improve the flow of sections. Tables are not standard, there is boarding information

6. Discussion: the discussion part is good, but you have to formulate the clinical and public health implications of the study findings. What are the innovative ideas, for scale up and ensure quality and safe services? Formulate clear what is innovating idea in the study.

• In study, only 23% of participants utilized all three ANC attributes adequately. This finding was not in line with the national standard,-------What was national figure? Is it possible to compare with the national data? You sad-----However, during this study, although the magnitude of adequate utilization was very low, --what is your assumption to say very low? It was moreover higher than the previous study done in Ethiopia only 2.6 % and 11% of participants received overall adequate ANC [24, 34].

7. Conclusion: Authors can suggest some recommendation, but not make a hard conclusion that those strategies would work or hinder.

8. Strengths and limitations of the study

• The use of tablets with digital survey tools for the data collection and ordinal regression analysis was the strength of this study. Do you think; is it the strength of study? On the other hand, due to the cross-sectional nature of the study, it is difficult to establish a temporal relationship between the determinants and outcome variables. It is already known facts, better to acknowledge other source of bias as a limitation of study.

Reviewer #2: Paper: Predictors of adequate antenatal care service utilization among mothers in Southern Ethiopia.

Assessment:

The study was designed to assess the magnitude and adequacy of antenatal care utilization in Southern Ethiopia. I am worried that the novelty of the study within the context of Ethiopia was not well defined. The authors cited several studies which show that poor utilization of antenatal care predisposes to poor maternal and neonatal outcomes in Ethiopia, and also report studies that feature the determinants of low utilization of antenatal care. While the study pinpoints the determinants of low demand for antenatal care, there is little efforts made to explain how supply factors feature in this relationship. As it is, the results of the study will find limited use in the design of actual policies and programs to improve the use and uptake of antenatal care in Ethiopia.

My major concern is the method section, where the authors defined the dependent variable of “adequate antenatal care utilization” as three composite attributes. The third attribute of “service content” was based on information obtained from the study participants, when indeed, it was clearly evident that some of the identified antenatal care components cannot be measured adequately by merely asking the participants. It was also difficult to identify what the standard measure of adequate antenatal care utilization was after using the 3 measures and how this was used in the eventual data analysis. The data analysis section is weak and does not seem to synchronize with the objectives and methodologies described in the paper.

My other concerns about the paper are listed below:

1) The paper is very poorly written with poor English language and disorderly presentation. Perhaps, an English language editor should edit the paper before it can be suitable for publication.

2) The introduction needs to be stronger and make a case to justify the novelty and justification for the study.

Reviewer #3: SUMMARY OF THE RESEARCH

1. The research sought to assess the level and predictors of adequate utilization of antenatal care (ANC) in the study setting. The importance of adequate ANC utilization – in mitigating the high level of maternal and neonatal mortality in many African countries – cannot be over emphasized.

2. The main claim of the report is that the prevalence of overall adequate ANC utilization, as defined by the authors, was generally low in the study setting.

3. The authors constructed the outcome variable ‘adequacy of ANC care’ from a composite of three variables (frequency of visits, timing and service content). This was a major strength considering that majority of previous studies assessed utilization of ANC only from the perspective of frequency of attendance.

4. A major weakness of this report is lack of clarity in the use of language making the article difficult to follow. I recommend that the authors should work with a native English speaker or copyeditor to improve the flow and readability of the text article.

5. Furthermore, the intention of the authors was to identify predictors of ‘overall adequate ANC utilization’, which is a discrete (and highest) category among the four (4) different categories of the outcome variable. They used ordinal logistic regression (OLR) towards achieving this. Ideally, OLR does not model the probability of a discrete category or outcome, it rather predicts the probability of increase (or decrease) across the different thresholds (or categories) of the ordinally ranked variable. Hence, the interpretation and presentation of the results with regards to predictors appears to be flawed, which is thus a major weakness of the manuscript.

SPECIFIC AREAS FOR IMPROVEMENT

MAJOR ISSUES

1) Introduction

a. Last paragraph: The justification, and indeed the entire paragraph, need to be re-written for clarity.

2) Methods

a. Source of data and data collection methods: Anderson Newman framework for healthcare utilization doesn’t have a questionnaire that can be adapted, the theory only provided some concepts that can be adapted. Hence, authors should rephrase the first statement under this sub heading.

b. Data quality control: The last paragraph appears cumbersome; what type of inconsistencies are the authors referring to in “line 6” of that paragraph, also what type of transformation did they do? How were the missing values mentioned in “line 7”managed? Is the questionnaire self or interviewer administered?

c. Authors need to provide further details regarding:

i. How the data was handled?

ii. The instrument used and how validity and reliability were ensured

d. Outcome/Predictor variables:

i. The outcome variable should simply be termed ‘ANC utilization’. This will remove the confusion between ‘ANC utilization’ as a variable and ‘adequate ANC utilization’ as one of the categories of that variable. Also, the four categories, as presented in the results, should be clearly defined in the methods.

ii. ‘Service contents’ was one of attributes used to construct the outcome variable. This presents a challenge with regards to the choice of explanatory or predictor variables (Table 1). The number of service contents received by a mother is essentially not within her domain of control. Such will depend on distal factors i.e. factors related to the health care providers and health care facilities. However, these variables were not included by the authors in their list of explanatory variables as contained in Table 1. This appears to be a major omission. It is recommended that, if the authors obtained any such data regarding health care provider / facility factors, such should be included as explanatory variables in the bivariate and multivariate analyses aimed at identifying predictors of ‘adequate ANC utilization’.

e. Data processing and analysis:

i. As stated by authors, all the assumptions for ordinal logistic regression (OLR) were considered. Were they met? If yes, it should be stated. What were the chi square statistics and p-value for the test of proportional odds (or test of parallel lines) for the model?

ii. The use of OLR poses some problems within the context of authors’ intention and the eventual interpretation of the results. The objective was to identify predictors of ‘overall adequate ANC utilization’, which is a discrete category, the highest among the four (4) different categories of the outcome variable. The OLR may not be the most appropriate for this purpose, even though the outcome variable appears to be ordinal in ranking. Ideally, OLR does not model the effect of explanatory variables on a discrete category or outcome, but rather their effect on the probability of increase (or decrease) across the different thresholds (or categories) of the ordinally ranked outcome variable. Two options are hereby recommended:

*a. Authors should revise the presentation of the results to reflect the proper interpretation of an OLR model. The reference category of the outcome variable must also be stated. In that case, the authors will be reporting the odds of obtaining higher levels of the categories of ANC utilization.

*b. Alternatively, to predict ‘overall adequate ANC utilization’ as a discrete category, there may be the need to dichotomize the outcome variable and apply the binary logistic regression. If authors prefer to retain the four categories, the use of multinomial regression should be considered.

3) Results

i. Knowledge and perception: How were these assessed, rated and/or categorized? The methods section provided no information about these variables.

ii. Levels of adequate utilization: The descriptive results of each of the 3 constituent variables i.e. frequency of visits, timing and service content, should be presented as a prelude to that of the composite ‘adequate ANC utilization’.

iii. Determinants of adequate utilization: As earlier stated, the interpretation of the results of OLR as presented here requires a major revision. It should be written in a manner that is easy for readers to follow. The following information should also be included: which variables had p<0.2 from bivariate analysis and thus included in multivariate model? Is table 3 showing all the variables included in the multivariate model or only those that turned out as significant predictors?

4) Grammar editing

i. The article will benefit greatly from grammar editing and this will help the readability of the article

MINOR ISSUES

1. Introduction

i. The literature was treated fairly however there is a document by WHO which may be of interest to the authors “Trends in maternal mortality 2000 to 2017: estimates by WHO, UNICEF, UNFPA, World Bank Group and the United Nations Population Division: executive summary” https://apps.who.int/iris/handle/10665/327596

2. Methods

i. Study design and setting: The population projection for Ethiopia as mentioned in line 4 should be properly referenced.

ii. Outcome variable – Service content: The authors should please clarify their statement “…over or up to mean (8) of 12…”

iii. Sampling procedure: The sentence in line 5, starting with Households were enrolled by systematic random sampling after identified eligible women…” is unclear, the authors should clarify to ensure that the readers understand what the authors did exactly.

3. Results

i. In paragraph 1 under result section, authors reported religion but this was not shown on table 2 as indicated

4. Discussion

i. A few of the literature used to discuss the findings are not similar with the results they are being compared with especially in the aspect of definition of utilization

5. Conclusion

i. The authors repeated results and made recommendations in the conclusion section rather than presenting a specific conclusion.

6. References

i. The authors should pay attention to the list of references and be consistent with the journal’s format for managing references.

6. PLOS authors have the option to publish the peer review history of their article (what does this mean?). If published, this will include your full peer review and any attached files.

Reviewer #1: No

Reviewer #2: **Yes: **Friday Okonofua

Reviewer #3: No

---

## [Author Response · Author response to Decision Letter 0]

20 Oct 2020

The comments and corrections forwarded from the reviewers and editor has been amended.The responses to the query raised by reviewers available in" response to reviewers" part .

---

## [Decision Letter · Decision Letter 1]

3 Mar 2021

PONE-D-20-07041R1

Magnitude and determinants of adequate antenatal care service utilization among mothers in Southern Ethiopia

PLOS ONE

Dear Dr. Gebrekirstos,

Thank you for submitting your manuscript to PLOS ONE. After careful consideration, we feel that it has merit but does not fully meet PLOS ONE’s publication criteria as it currently stands. Therefore, we invite you to submit a revised version of the manuscript that addresses the points raised during the review process.

In the revised version, you have incorporated most of the comments of reviewers/editor. Nevertheless, I will suggest including figures from the table in the text of findings section on line 252-268. 

We look forward to receiving your revised manuscript.

Kind regards,

Rubeena Zakar, Ph.D

Academic Editor

PLOS ONE

Journal Requirements:

Reviewers' comments:

Reviewer's Responses to Questions

**Comments to the Author**

1. If the authors have adequately addressed your comments raised in a previous round of review and you feel that this manuscript is now acceptable for publication, you may indicate that here to bypass the “Comments to the Author” section, enter your conflict of interest statement in the “Confidential to Editor” section, and submit your "Accept" recommendation.

Reviewer #1: (No Response)

Reviewer #3: (No Response)

2. Is the manuscript technically sound, and do the data support the conclusions?

Reviewer #1: Yes

Reviewer #3: Yes

3. Has the statistical analysis been performed appropriately and rigorously? 

Reviewer #1: Yes

Reviewer #3: Yes

4. Have the authors made all data underlying the findings in their manuscript fully available?

Reviewer #1: Yes

Reviewer #3: Yes

5. Is the manuscript presented in an intelligible fashion and written in standard English?

Reviewer #1: Yes

Reviewer #3: No

6. Review Comments to the Author

Reviewer #1: Topic: Magnitude and determinants of adequate antenatal care service utilization among mothers in Southern Ethiopia;

Version 2

General comments

Abstract

1. Result: better to narrate the finding line with the objective of the study

2. Conclusion: better to recommend specific action

3. Authors extensively edited the manuscript and there is great improvement. Authors responded the comments point by point, but some of the comments are not well addressed as it stated.

Reviewer #3: The Authors have addressed the issues raised earlier. However, they should further attend to the following;

1. On table 5, the abbreviation that were listed under the key were not presented on the table. Please Check and correct appropriately

2. Check the spelling of category under the key, it is written as "catagory"

2. Grammar editing: there is still more to be done, I don't think Grammarly alone is sufficient here, the authors should please get a native speaker to help proof read the manuscript.

7. PLOS authors have the option to publish the peer review history of their article (what does this mean?). If published, this will include your full peer review and any attached files.

Reviewer #1: No

Reviewer #3: No

---

## [Author Response · Author response to Decision Letter 1]

12 Mar 2021

Manuscript Title: Magnitude and determinants of antenatal care service utilization among mothers in Southern Ethiopia 2019. 

Dear editor and reviewers 

We would like to extend our heartfelt gratitude and appreciation for your valuable comments and priceless time. Thank you, your comments have helped us a lot to improve the manuscript. We tried to address all comments point by point in this paper. 

Responses for the editor 

1. In the revised version, you have incorporated most of the comments of reviewers/editor. Nevertheless, I will suggest including figures from the table in the text of findings section on line 252-268

Response: Comment accepted and the manuscript has been revised including figures from the table, please see the highlighted manuscript from line (271-316)

Response: No retracted manuscript was cited. But some changes have been made to the reference, because there were mistakenly cited references. So, we made replacement to some references, please see the highlighted manuscript from line (577-604) 

• We also remove two references because of redundancy, please see the highlighted manuscript line 651 and 680

Response to reviewer #1

Abstract

1. Result: better to narrate the finding line with the objective of the study

Response: comment accepted and amendment done, please see the highlighted manuscript from line (34-42)

2. Conclusion: better to recommend specific action

 Response: comment accepted and amendment done, please see the highlighted manuscript from line (43-46)

3. Authors extensively edited the manuscript and there is great improvement. Authors responded the comments point by point, but some of the comments are not well addressed as it stated.

Response: Thank you for your comment; we tried to revise the whole revised manuscript line by line to address all the comments given

Response to reviewer #3 

1. On table 5, the abbreviations that were listed under the key were not presented on the table. Please Check and correct appropriately

Response: comment accepted and amendment done. Please see the highlighted manuscript from line (747-749)

2. Check the spelling of category under the key, it is written as "catagory"

Response: Comment accepted and amendment done. Please see the highlighted manuscript line 747

3. Grammar editing: there is still more to be done, I don't think Grammarly alone is sufficient here, the authors should please get a native speaker to help proof read the manuscript.

Response: Comment accepted and amendment has been done. We tried to improve the readability of the manuscript by using native speaker.

---

## [Decision Letter · Decision Letter 2]

28 Apr 2021

Magnitude and determinants of adequate antenatal care service utilization among mothers in Southern Ethiopia

PONE-D-20-07041R2

Dear Dr. GEBREKIRSTOS,

We’re pleased to inform you that your manuscript has been judged scientifically suitable for publication and will be formally accepted for publication once it meets all outstanding technical requirements.

Kind regards,

Tanya Doherty, PhD

Academic Editor

PLOS ONE

Additional Editor Comments (optional):

Please do a thorough English language edit of the revised manuscript as there remain grammatical errors and typo's.

Reviewers' comments:

Reviewer's Responses to Questions

**Comments to the Author**

1. If the authors have adequately addressed your comments raised in a previous round of review and you feel that this manuscript is now acceptable for publication, you may indicate that here to bypass the “Comments to the Author” section, enter your conflict of interest statement in the “Confidential to Editor” section, and submit your "Accept" recommendation.

Reviewer #1: All comments have been addressed

2. Is the manuscript technically sound, and do the data support the conclusions?

Reviewer #1: Yes

3. Has the statistical analysis been performed appropriately and rigorously? 

Reviewer #1: Yes

4. Have the authors made all data underlying the findings in their manuscript fully available?

Reviewer #1: Yes

5. Is the manuscript presented in an intelligible fashion and written in standard English?

Reviewer #1: Yes

6. Review Comments to the Author

Reviewer #1: (No Response)

7. PLOS authors have the option to publish the peer review history of their article (what does this mean?). If published, this will include your full peer review and any attached files.

Reviewer #1: **Yes: **Akine Eshete Abosetugn

---

## [Editor Report · Acceptance letter]

5 May 2021

PONE-D-20-07041R2 

Magnitude and determinants of adequate antenatal care service utilization among mothers in Southern Ethiopia 

Dear Dr. Gebrekirstos :

I'm pleased to inform you that your manuscript has been deemed suitable for publication in PLOS ONE. Congratulations! Your manuscript is now with our production department. 

Kind regards, 

on behalf of

Professor Tanya Doherty 

Academic Editor

PLOS ONE